# Variants associating with uterine leiomyoma highlight genetic background shared by various cancers and hormone-related traits

Thorunn Rafnar [1], Bjarni Gunnarsson[1], Olafur A. Stefansson[1], Patrick Sulem [1], Andres Ingason[1],
Michael L. Frigge[1], Lilja Stefansdottir[1], Jon K. Sigurdsson[1], Vinicius Tragante[1,2], Valgerdur Steinthorsdottir[1],
Unnur Styrkarsdottir [1], Simon N. Stacey[1], Julius Gudmundsson[1], Gudny A. Arnadottir [1],
Asmundur Oddsson [1], Florian Zink[1], Gisli Halldorsson [1], Gardar Sveinbjornsson[1], Ragnar P. Kristjansson[1],
Olafur B. Davidsson[1], Anna Salvarsdottir[3], Asgeir Thoroddsen[3], Elisabet A. Helgadottir[3], Katrin Kristjansdottir[3],
Orri Ingthorsson[4], Valur Gudmundsson[4], Reynir T. Geirsson[3,5], Ragnheidur Arnadottir[3],
Daniel F. Gudbjartsson [1,6], Gisli Masson[1], Folkert W. Asselbergs[2,7,8,9], Jon G. Jonasson[5,10], Karl Olafsson[3],
Unnur Thorsteinsdottir[1,5], Bjarni V. Halldorsson [1,11], Gudmar Thorleifsson[1] & Kari Stefansson[1,5]

Uterine leiomyomas are common benign tumors of the myometrium. We performed a meta-analysis of two genome-wide association studies of leiomyoma in European women (16,595 cases and 523,330 controls), uncovering 21 variants at 16 loci that associate with the disease. Five variants were previously reported to confer risk of various malignant or benign tumors (rs78378222 in *TP53*, rs10069690 in *TERT*, rs1800057 and rs1801516 in *ATM*, and rs7907606 at *OBFC1*) and four signals are located at established risk loci for hormone-related traits (endometriosis and breast cancer) at 1q36.12 (*CDC42/WNT4*), 2p25.1 (*GREB1*), 20p12.3 (*MCM8*), and 6q26.2 (*SYNE1/ESR1*). Polygenic score for leiomyoma, computed using UKB data, is significantly correlated with risk of cancer in the Icelandic population. Functional annotation suggests that the non-coding risk variants affect multiple genes, including *ESR1*. Our results provide insights into the genetic background of leiomyoma that are shared by other benign and malignant tumors and highlight the role of hormones in leiomyoma growth.

[1] deCODE Genetics/Amgen, Sturlugata 8, 101 Reykjavik, Iceland. [2] Department of Cardiology, Division Heart & Lungs, University Medical Center Utrecht, University of Utrecht, 3584 CX Utrecht, The Netherlands. [3] Department of Obstetrics and Gynecology, Landspitali University Hospital, 101 Reykjavik, Iceland. [4] Department of Obstetrics and Gynecology, Akureyri Hospital, 600 Akureyri, Iceland. [5] Faculty of Medicine, School of Health Sciences, University of Iceland, 101 Reykjavik, Iceland. [6] School of Engineering and Natural Sciences, University of Iceland, 101 Reykjavik, Iceland. [7] Durrer Center for Cardiovascular Research, Netherlands Heart Institute, 3501 DG Utrecht, The Netherlands. [8] Institute of Cardiovascular Science, Faculty of Population Health Sciences, University College London, London WC1E 6HX, UK. [9] Farr Institute of Health Informatics Research and Institute of Health Informatics, University College London, London NW1 2DA, UK. [10] Department of Pathology, Landspitali University Hospital, 101 Reykjavik, Iceland. [11] School of Science and Engineering, Reykjavik University, 101 Reykjavik, Iceland. These authors contributed equally: Thorunn Rafnar, Bjarni Gunnarsson. Correspondence and requests for materials should be addressed to T.R. (email: thorunn.rafnar@decode.is) or to K.S. (email: kstefans@decode.is)

Uterine leiomyomas, sometimes referred to as fibroids, are benign tumors that arise in the smooth muscle cells of the uterine wall[1]. Leiomyomas represent the most common pelvic tumors in women with prevalence by the age 50, above 80% for African–American women, and nearly 70% for Caucasian women[2]. Although leiomyomas are non-malignant and in majority of cases not symptomatic, about 20–25% of cases cause problems that warrant treatment. These symptoms include dysmenorrhea, pelvic pain, abnormal bleeding, infertility, and complications during pregnancy[3]. Leiomyomas are estrogen responsive and many of the non-surgical therapeutic options involve some form of hormonal manipulation; however, in many cases, the tumors recur when therapy is discontinued[1]. Leiomyomas are the most common indication for hysterectomy and the second most common cause of surgical intervention for women after cesarean section, with over 600,000 hysterectomies performed per year in the USA[4].

The biological mechanisms leading to the development of uterine leiomyomas are not well understood. Risk factors include age, obesity, increased levels of estrogens, hypertension, parity, and race, with prevalence in African–American women being considerably higher than that in European–American women[2]. Genetic factors are also implicated in leiomyoma formation: First-degree relatives of affected women have a 2.5-fold greater risk of developing the condition than the population average[5] and the concordance among monozygotic twins is almost twice that of dizygotic twins[6]. A genome-wide association study (GWAS) in Japanese women found variants at three loci that associate with uterine leiomyoma, at 22q13.1 (*TNRC6B*), 11p15.5 (*BET1L*), and 10q24.33 (*OBFC1*)[7]. The associations between the disease and the

first two loci have been confirmed in European women[8]. Finally, a GWAS of leiomyoma risk among African–American women yielded a locus on 22q13.1 (*CYTH4*)[9].

To gain insights into the genetic causes of uterine leiomyomas, we performed a meta-analysis of leiomyoma in Europeans, using GWAS data on 16,595 cases and 523,330 controls from Iceland and the UK, identifying 21 variants at 16 loci that associate with the disease.

## Results

**Association analysis.** We combined the results of two GWAS' of uterine leiomyoma, one from Iceland and the other from the UK Biobank (UKB), using a total of 16,595 cases and 523,330 controls of confirmed Europen descent. The Icelandic GWAS consists of 6728 hospital-based, histologically confirmed leiomyoma cases and 124,542 controls, whereas the UKB leiomyoma dataset is based on 9867 hospital-based cases and 398,788 controls. A Manhattan plot of the meta-analysis results is shown in Fig. 1. As the probability of a variant impacting a phenotype differs between the functional annotation classes[10], we applied genome-wide significance thresholds using a weighted Bonferroni procedure that takes this prior probability into account and corrects for all 43,421,991 variants tested (Supplementary Data 1, Methods). Using this scheme, a total of 412 variants at 16 loci reach the threshold of genome-wide significance (Supplementary Data 2). We applied conditional analysis to search for distinct signals at each locus and find additional variants at five loci that associate with leiomyoma independent of the main signal (with $P_{cond} < 10^{-6}$); three of the secondary signals reach genome-wide

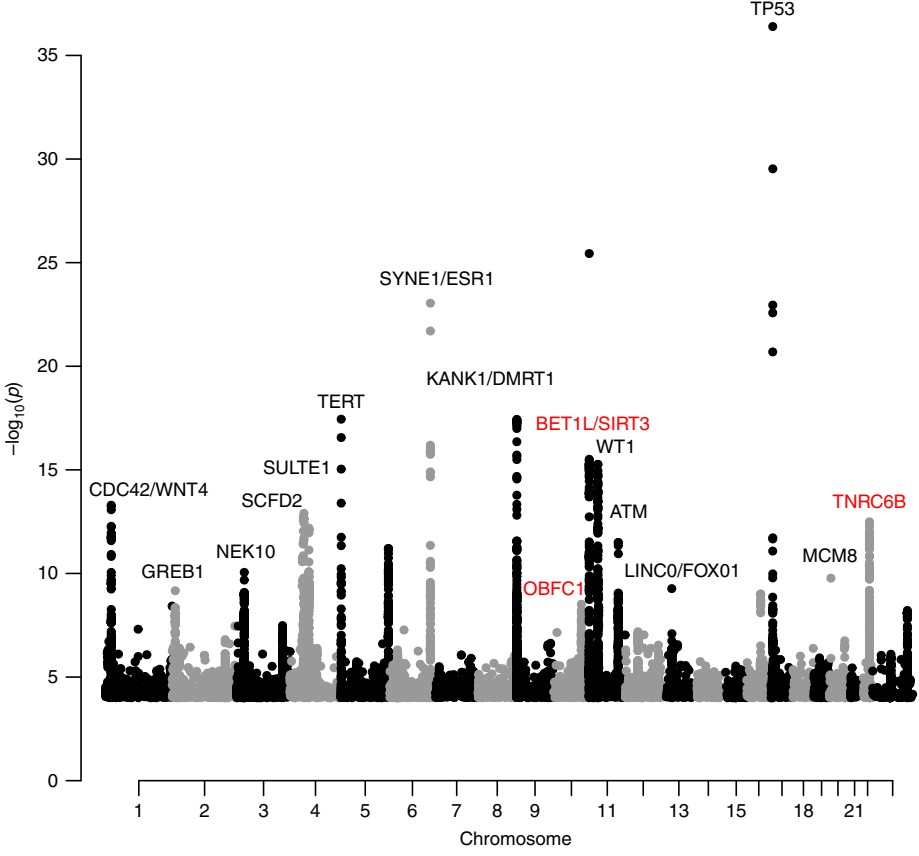

**Fig. 1** Manhattan plot of the association results of meta-analysis of leiomyoma of the uterus. The Manhattan plot shows variants with *P*-value < 1.0 × 10$^{-3}$ in a meta-analysis of GWAS data from 20,621 leiomyoma patients and 280,541 controls of European ancestry. Shown are negative log10-transformed *P* values (*y*-axis) over 22 autosomes (*x*-axis). Red font color indicates leiomyoma loci previously reported in the Japanese population

**Table 1 Association results for lead variants at loci reaching genome-wide significance in meta-analysis of leiomyoma**

| Marker Name Pos hg38[1] | coding effect | EAF | EA/OA | Gene | OR$^2_{meta}$ (95% CI) | P$_{meta}$ | Phenotypes reported at locus[3] |
|---|---|---|---|---|---|---|---|
| rs10917151 chr1:22096228 | downstream | 0.201 | A/G | CDC42/WNT4 | 1.12 (1.09, 1.16) | 5.1E-14 | Endometriosis, BMD, OvCa |
| rs148143917 chr2:11524625 | upstream | 0.019 | C/A | GREB1 | 0.74 (0.67, 0.82) | 8.1E-10 | Endometriosis, BrCa |
| rs10929757 chr2:11562535 | missense Asn77Thr | 0.410 | A/C | GREB1 | 0.92 (0.90, 0.94) | 8.1E-12 | Endometriosis, BrCa |
| rs479404 chr3:27321573 | intron | 0.322 | C/T | NEK10 | 1.09 (1.06, 1.12) | 8.9E-11 | BrCa |
| rs765333492 chr4:53021103 | intron | 0.003 | C/T | SCFD2 | 2.76 (2.11, 3.6) | 2.3E-13 | |
| rs2202282 chr4:69768723 | intergenic | 0.485 | T/C | SULT1E1 | 1.09 (1.07-1.12) | 6.5E-13 | |
| rs10069690 chr5:1279675 | intron | 0.259 | T/C | TERT | 1.12 (1.1, 1.15) | 3.6E-18 | ThCa, BrCa, CLL, TeCa, PrCa, UBC, PaCa, glioma |
| rs58415480 chr6:152241136 | intron | 0.142 | G/C | SYNE1/ESR1 | 1.18 (1.14, 1.22) | 9.0E-24 | Endometriosis |
| rs73639400 chr9:683423 | intron | 0.142 | C/T | KANK1 | 1.12 (1.09-1.16) | 7.9E-12 | |
| rs7030354 chr9:804231 | intergenic | 0.404 | T/C | DMRT1 | 1.11 (1.08, 1.14) | 2.0E-17 | |
| rs7907606 chr10:103920874 | upstream | 0.155 | G/T | OBFC1 | 1.1 (1.07, 1.14) | 3.1E-09 | LM (Japan), BCC, multiple cancers, telomere length |
| rs11246001 chr11:210899 | upstream | 0.044 | T/C | BET1L | 0.82 (0.78-0.87) | 5.2E-12 | LM (Japan) |
| rs507139 chr11:225196 | intron | 0.082 | A/G | SIRT3 | 0.83 (0.80, 0.87) | 1.4E-16 | LM (Japan) |
| rs11031731 chr11:32343884 | intergenic | 0.158 | A/G | WT1 | 1.14 (1.10-1.17) | 5.4E-16 | |
| rs1800057 chr11:108272729 | missense Pro1054Arg | 0.020 | G/C | ATM | 1.28 (1.19, 1.38) | 3.1E-11 | CLL, PrCa, RCC |
| rs1801516[4] chr11:108304735 | missense Asp1853Asn | 0.152 | A/G | ATM | 0.91 (0.88, 0.94) | 7.4E-08 | CMM, response to radio/chemotherapy |
| rs117245733 chr13:40149807 | intergenic | 0.020 | A/G | LINC0 | 1.32 (1.21, 1.44) | 2.2E-10 | |
| rs79864407[4] chr13:40605661 | intron | 0.305 | G/A | FOXO1 | 1.07 1.04, 1.10() | 7.0E-08 | |
| rs78378222 chr17:7668434 | 3'UTR | 0.018 | G/T | TP53 | 1.74 (1.6, 1.89) | 4.0E-37 | BCC, PrCa, glioma, CR adenoma, NB |
| rs16991615 chr20:5967581 | missense Glu341Lys | 0.081 | A/G | MCM8 | 1.16 (1.11, 1.21) | 3.6E-10 | Age at menopause, BrCa |
| rs12484951 chr22:40307071 | intron | 0.252 | G/T | TNRC6B | 1.11 (1.08, 1.14) | 3.2E-13 | LM (Japan) |

See **Supplementary Data 6** for results in the Icelandic and UKB datasets and references for the reported signals referred to in the table
[1]Marker positions are according to GRCh38/hg38,
[2]Odds-ratios correspond to effect alleles,
[3]Phenotypes relating to tumorigenesis or hormone-related traits previously reported at the locus,
[4]Secondary signals reaching conditional P value < 10-6 (approximately 30.000 variants tested)
Abbreviations: EAF, effect allele frequency; EA, effect allele; OA, other allele; P$_{Meta}$, P-value for fixed effects meta-analysis; OR$_{Meta}$, Odds ratio; BMD, bone mineral density; OvCa, ovarian cancer; BrCa, breast cancer; TeCa, testicular cancer; LM, leiomyoma, PrCa, prostate cancer; UBC, urinary bladder cancer; ThCa, thyroid cancer; PaCa, pancreatic cancer; BCC, basal cell carcinoma, CLL, chronic lymphocytic leukemia: RCC, renal cell carcinoma; CMM, cutaneous malignant melanoma; CR adenoma, colorectal adenoma; NB, neuroblastoma

significance. The 19 genome-wide significant markers and two significant secondary signals are listed in Table 1 and Supplementary Data 3, locus plots for all the markers are presented in Supplementary Fig. 1, and all association signals with P value < 5.0 × 10⁻⁸ are listed in Supplementary Data 4. We also tested 21 markers for association with 20 different tumor types using information on 42,331 cancer cases from the Icelandic Cancer Registry and several phenotypes that are strongly affected by estrogen exposure, i.e., endometriosis (1,857 cases), bone mineral density (BMD, 28,900 individuals), and age at menopause (10,216 individuals) (Supplementary Data 5 and 6).

**Variants at 16 loci associate with leiomyoma.** Examination of the meta-analysis results suggests that at least two genetic pathways play a role in the development of leiomyomas; one linked to tumorigenesis and the other linked to hormone-related traits (Table 1, Supplementary Data 3). The most significant association with leiomyoma is with a low-frequency 3'UTR variant in TP53, rs7837822_G (P = 4.03 × 10⁻³⁷, meta-analysis of logistic regression, OR = 1.74). The same allele of this variant was previously reported to associate with increased risk of several malignant and benign tumor types, including basal cell skin cancer, glioma, prostate cancer, neuroblastoma, and colorectal adenoma[11–13]. Four additional leiomyoma variants in our meta-analysis, one at the TERT locus, two in ATM, and one at the OBFC1 locus, have previously been associated with cancer risk. At the TERT locus, rs10069690_T was previously reported to increase the risk of thyroid cancer[14], estrogen and progesterone receptor-negative breast cancer[15,16], CLL[17], and glioma[18] and

decrease the risk of testicular, prostate, bladder, and pancreatic cancers[18]. Two uncorrelated ($r^2 = 0.007$) missense variants in the *ATM* gene, rs1801516 (Asp1853Asn) and rs1800057 (Pro1054Arg), independently associate with leiomyoma. rs1801516_A was previously reported to associate with decreased melanoma risk[19] and response to chemotherapy and radiotherapy[20,21]. This variant shows significant association with increased risk of squamous cell skin cancer in Iceland ($P = 4.35 \times 10^{-5}$, logistic regression, OR = 1.21) after correction for the number of phenotypes tested ($P < 0.05/29 = 1.72 \times 10^{-3}$), the direction of effect being opposite to the reported melanoma effect (Supplementary Data 6). The other leiomyoma missense variant in *ATM*, rs1800057_G, has been linked to increased risk of chronic lymphocytic leukemia[22], prostate cancer[23], and renal cell carcinoma[24]. Finally, at the *OBFC1* locus, the strongest leiomyoma risk variant, rs7907606_G, has previously been associated with the risk of basal cell carcinoma[25] and is strongly correlated with variants that have been reported for risk of melanoma[26], adenocarcinoma of the lung[27], thyroid cancer[14], renal cell carcinoma[24], serous ovarian cancer[28], glioma[29], and CLL[30], as well as to telomere length[31].

Several of the leiomyoma variants and loci emerging from our meta-analysis have been linked to hormone-related traits. Variants at the *CDC4/WNT4* locus have been associated with endometriosis, BMD, bone size, and ovarian cancer[32–35]. The leiomyoma variant rs10917151 at this locus is correlated with the strongest reported endometriosis variant ($r^2 = 1$), as well as the ovarian cancer variant ($r^2 = 0.97$), and one of the BMD variants ($r^2 = 0.81$) (Table 1, Supplementary Data 3). The missense variant in *MCM8*, rs16991615_A, that associates with increased risk of leiomyoma in our study, also associates with delayed onset of menopause[36], breast cancer[37], and with increased BMD in Iceland (Supplementary Data 6). At the *SYNE1/ESR1* locus, the strongest leiomyoma variant, rs58415480, is correlated with a reported endometriosis variant rs71575922 ($r^2 = 0.94$)[32]. Notably, the leiomyoma variant is not correlated to any of the six independent breast cancer signals reported in European populations at this locus, represented by the variants rs9397435/rs9397437, rs3757322, rs851984, rs9918437, rs78796841, and rs2747652[37–39] (all $r^2 < 0.03$ in CEU). Two variants at the *GREB1* locus, the missense variant rs10929757 (Asn77Thr) and rs148143917 (upstream variant), associate with leiomyoma. Two association signals for endometriosis have been reported at this locus[32], but neither are strongly correlated with the leiomyoma variants (Table 1). The leiomyoma variant rs10929757 (Asn77Thr) associates with endometriosis, whereas only one of the endometriosis variants, rs77294520, associates with leiomyoma in the Icelandic data (Supplementary Data 6 and 7). Two additional variants have potential links to hormone metabolism. First, rs2202282 at 4q13.3 is located in a cluster of sulfotransferase genes, including *SULT1E1*, which catalyzes the sulfation of estrogens[40]. Second, the low-frequency variant rs765333492 (*SCFD2*, MAF = 0.3%) found only in the Icelandic dataset also shows suggestive association with endometriosis in Iceland ($P = 0.018$, logistic regression, OR = 1.92) (Supplementary Data 6).

Variants at three loci previously reported for uterine fibroids in Japanese women[7], 22q13.1 (*TNRC6B*), 11p15.5 (*BET1L*), and 10q24.33 (*OBFC1*), reached genome-wide significance in our meta-analysis. Our top *TNCRB6* variant is strongly correlated with the variant reported in Japanese women ($r^2 = 0.97$). At the *BET1L* locus, we found two distinct signals, one of which is fully correlated with the reported *BET1L* variant in Japanese women (Table 1, Supplementary Data 3). At the *OBFC1* locus, the previously discussed cancer-associated variant rs7907606 is not correlated with the Japanese leiomyoma variant reported (rs7913069, $P = 0.98$, meta-analysis of logistic regression, OR =

1.0, $r^2 = 0.001$ in Europeans and 0.014 in Japanese). To further analyze the Japanese signal in our data, we tested 278 variants that correlate with rs7913069 (with $r^2 > 0.1$) in the Japanese population for association with leiomyoma in our dataset (Supplementary Data 8). The lowest P-value observed is 0.00015 for rs78381949, a variant that correlates with $r^2 = 0.23$ with rs7913069 in East-Asians, but has very little correlation with rs7907606 ($r^2 = 0.04$) in Caucasians. It is thus unlikely that rs7907606 and rs7913069 are tagging the same association signal. Finally, the variant at the *CYTH4* locus identified in African–American women[9], rs739187, does not associate with leiomyoma in our data ($P = 0.51$, meta-analysis of logistic regression, OR = 1.01). rs739187 is about 3 Mb away from the leiomyoma variant rs12484951 (*TNRC6B*) and the 2 variants are not correlated either in Europeans ($r^2 = 0.0064$) or African–Americans ($r^2 = 0.017$).

**Overlap of leiomyoma, endometriosis, and endometrial cancer.** Although a large fraction of hysterectomies are due to leiomyoma, hysterectomies performed for other clinical conditions such as endometriosis, uterine prolapse, and cancer may also lead to diagnosis of leiomyoma. We assessed whether the leiomyoma signals are driven by endometriosis cases within the leiomyoma group. To this end, we removed all known endometriosis cases from the Icelandic leiomyoma sample set and compared the association results with the results obtained for the full Icelandic leiomyoma group. The effects of leiomyoma associations were not diminished for any of the variants (Supplementary Data 6). We note that since we only have information on surgically diagnosed endometriosis cases, it is possible that there may be some undiagnosed cases left in the leiomyoma group. We also assessed the association of all 19 reported endometriosis variants[32] with the endometriosis-excluded leiomyoma group. Only rs12037376 (*CDC42/WNT4*), rs71575922 (*SYNE1/ESR1*), and rs77294520 (*GREB1*) associate with leiomyoma after correction for the number of tests ($P < 0.05/19 = 2.6 \times 10^{-3}$), with all three variants showing the same direction of effect for endometriosis and leiomyoma (logistic regression, Supplementary Data 7).

We tested if the leiomyoma variants associate with endometrial cancer, using 2462 endometrial cancer cases (1009 from Iceland and 1453 from UKB) and 336,978 controls (218,565 from Iceland and 118,413 from UKB) (Supplementary Data 9). Only rs10917151 (CDC42/WNT4) associates with endometrial cancer ($P = 4.5 \times 10^{-4}$, logistic regression, OR 1.14) after correcting for the number of tests ($P = 0.05/21 = 0.0024$). We also tested the association between leiomyoma and seven variants reported in GWAS studies of endometrial cancer[41]. None of the endometrial cancer variants associate with leiomyoma (Supplementary Data 10).

**Functional annotation of non-coding leiomyoma variants.** To assess the functional role of the non-coding leiomyoma variants, we annotated the lead non-coding variants along with correlated variants ($r^2 > 0.8$, $N = 438$), using data on regulatory regions in uterine tissue from the Encode project (www.encodeproject.org). Specifically, we focused on acetylation of lysine K27 of histone H3 (H3K27ac) and open chromatin regions (DNase Hypersensitivity Sites; DHS). This analysis identified 51 out of 438 variants that intersected with H3K27ac marked regions (Supplementary Data 11). In addition, 14 variants were found within DHS sites, of which 10 are also within H3K27ac regions. To search for gene targets of variants in regulatory regions, we analyzed chromatin interaction maps and identified 85 potential target genes that associate with one or more of the candidate regulatory variants (Supplementary Data 11, Fig. 2, Supplementary Fig. 2). Genes

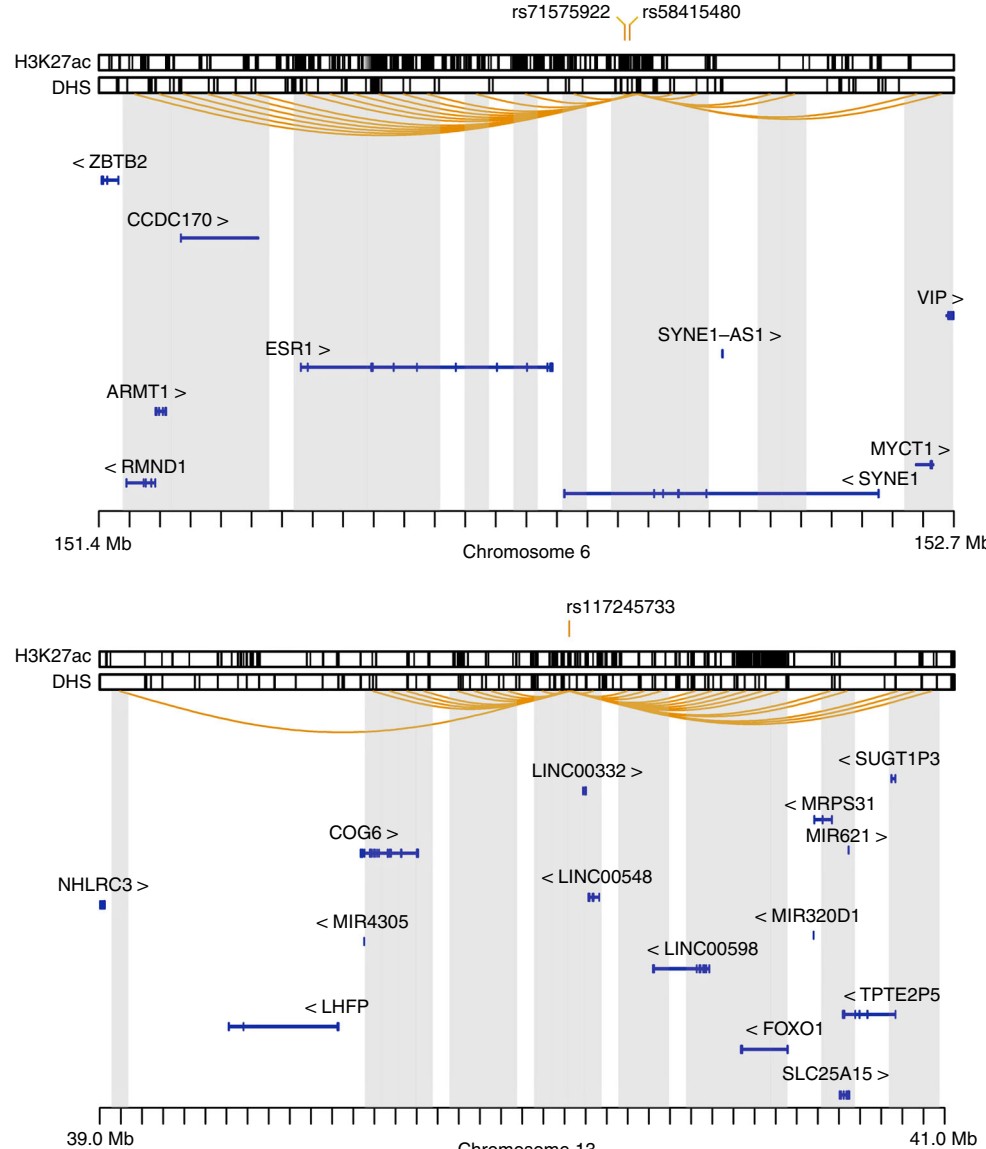

**Fig. 2** Leiomyoma risk variants intersecting with regulatory regions and their candidate target genes. Two loci on chromosomes 6 (upper panel) and 13 (lower panel) are shown as examples. Acetylation of lysine residue K27 of histone H3 (H3K27ac) and open chromatin (DHS, DNase Hypersensitivity Sites) in the uterus samples are shown as tick marks at the top. At chromosome 6, the lead variant rs58415480 is shown along with the only other variant in the same LD class ($r^2 > 0.8$), rs71575922. At chromosome 13, rs117245733 is in an LD class by itself, i.e., does not have strongly correlated markers. Candidate target genes were identified by the analysis of available chromatin interaction maps, shown on the figure as orange arcs. The shaded regions represent regions found in contact with the regulatory variants

whose transcription start sites (TSS) are found within 1 kb of a regulatory variant are listed (Supplementary Data 11).

The variant rs58415480 at 6q25.2 (*SYN1/ESR1*) is in strong LD with only one other variant, rs71575922 ($r^2 = 0.96$) and both reside in a regulatory region in the uterine tissue. According to chromosomal interaction data, rs58415480 and rs71575922 are candidate regulators of seven target genes including *ESR1*, which is important in the growth of leiomyomas (Fig. 2, upper panel). The variant on chromosome 13, rs117245733, does not have a strongly correlated marker. This variant resides in a region densely marked by H3K27ac in the uterine tissue, indicative of regulatory activity (Fig. 2, lower panel). Out of the eleven candidate target genes regulated by rs117245733, the *FOXO1* gene is notable, as its expression in primary human endometrial stromal cells is regulated by *ESR1*[42] and it is a direct protein–protein interaction partner of ESR1 in breast cancer

cells[43]. The secondary signal at 13q14.11, rs7986407, also targets *FOXO1* (Supplementary Fig. 2). At 11p13, the well-known tumor suppressor gene *WT1* interacts with several variants, with rs10835884, rs11031716, and rs11828433 representing the top-ranking causal variants (Supplementary Data 11, Supplementary Fig. 2). According to the Human Protein Atlas (www.proteinatlas. org), *WT1* expression is limited to smooth muscle cells of the uterus, in addition to Fallopian tube, testis, and kidney tissue.

**Heritability and polygenic score (PGS)**. We used LD score regression[44] to estimate the SNP heritability of leiomyoma. Using the Icelandic population, we estimate the SNP heritability to be 13% (95% CI 4–22%). We estimate the sibling recurrence risk in Iceland to be 1.74 (95% CI 1.63, 1.85, $P < 0.0001$) and find that the proportion of sibling recurrence risk for leiomyoma explained by the 21 variants is 10.8%.

**Table 2 Polygenic risk score analysis using chip-genotyped cases and controls in Iceland**

| Case-control phenotypes | N aff | N ctrl | R² [%] | P | OR (95% CI) |
|---|---|---|---|---|---|
| Leiomyoma of Uterus | 5234 | 123,894 | 0.60% | 3.2E-55 | 1.25 (1.22 1.29) |
| Leiomyoma wo endometriosis | 4876 | 124,252 | 0.58% | 1.4E-50 | 1.25 (1.21 1.29) |
| All Cancer | 19,712 | 124,522 | 0.02% | 2.8E-06 | 1.04 (1.02 1.06) |
| Thyroid Cancer | 696 | 136,207 | 0.21% | 3.0E-05 | 1.17 (1.09 1.26) |
| Prostate Cancer | 3434 | 100,757 | 0.03% | 1.7E-03 | 1.06 (1.02 1.1) |
| Kidney Cancer | 734 | 112,893 | 0.10% | 2.4E-03 | 1.12 (1.04 1.2) |
| Endometriosis | 1385 | 134,434 | 0.05% | 3.4E-03 | 1.08 (1.02 1.14) |
| Endometriosis Stage III and IV | 542 | 127,502 | 0.11% | 4.1E-03 | 1.13 (1.04 1.23) |
| Colorectal adenoma | 8735 | 128,569 | 0.01% | 7.5E-03 | 1.03 (1.01 1.05) |
| MGUS | 978 | 121,348 | 0.04% | 0.022 | 1.08 (1.01 1.15) |
| BCC | 3948 | 125,581 | 0.01% | 0.036 | 1.04 (1.01 1.07) |
| Cervical Cancer | 377 | 136,526 | 0.07% | 0.049 | 1.11 (1 1.23) |
| Melanoma | 1281 | 141,744 | 0.01% | 0.16 | 1.04 (0.98 1.1) |
| Bladder Cancer | 1106 | 136,208 | 0.01% | 0.20 | 0.961 (0.9 1.02) |
| Breast Cancer | 3685 | 140,325 | 0.00% | 0.27 | 1.02 (0.99 1.06) |
| UADT Cancer | 268 | 113,299 | 0.02% | 0.32 | 0.941 (0.83 1.06) |
| Leiomyosarcoma | 22 | 96,914 | 0.26% | 0.36 | 1.22 (0.8 1.85) |
| SQCSC | 1452 | 135,852 | 0.00% | 0.43 | 0.979 (0.93 1.03) |
| Gastric Cancer | 411 | 93,609 | 0.00% | 0.58 | 1.03 (0.93 1.14) |
| NH Lymphoma | 558 | 142,467 | 0.00% | 0.66 | 1.02 (0.94 1.11) |
| Brain Cancer | 172 | 142,343 | 0.01% | 0.69 | 1.03 (0.89 1.2) |
| Pancreatic Cancer | 324 | 103,797 | 0.00% | 0.74 | 0.981 (0.88 1.1) |
| Ovarian Cancer | 280 | 141,320 | 0.00% | 0.88 | 1.01 (0.9 1.14) |
| Endometrial Cancer | 564 | 121,762 | 0.00% | 0.94 | 0.997 (0.92 1.08) |
| Lung Cancer | 1972 | 120,354 | 0.00% | 0.95 | 0.998 (0.95 1.04) |
| Colorectal Cancer | 2001 | 127,528 | 0.00% | 0.95 | 0.998 (0.95 1.04) |
| **Quanitative trait phenotypes** | **N aff** | **N ctrl** | **R²** | **P** | **Beta** |
| Age at Menopause | 9719 | NA | 1.00E−04 | 0.20 | 0.049 (−0.03 0.12) |
| BMD Spine | 22,059 | NA | 0.00% | 0.38 | 0.006 (−0.01 0.02) |
| BMD Hip | 24,297 | NA | 0.00% | 0.86 | 0.001 (−0.01 0.01) |

$R^2$ denotes explained variance, OR is the odds ratio for binary phenotypes, and Beta is the effect for qtl phenotypes
*MGUS* monoclonal gammopathy of undetermined significance, *BCC* basal cell carcinoma of the skin, *UADT* upper aero-digestive tract, *SQCSC* squamous cell skin cancer, *BMD* bone mineral density

The sharing of susceptibility loci of leiomyoma with different tumor types on one hand and hormone-related traits on the other suggests that leiomyomas are genetically correlated with these phenotype groups. We used a polygenic score (PGS) to map the genetic correlation between leiomyoma and 28 selected tumor/hormone-related phenotypes (Table 2). To avoid confounding, risk alleles, $P$-values, and effect estimates were extracted from the UKB study on leiomyoma and used to calculate a standardized PGS for the genotyped cases in the Icelandic dataset. As expected, the PGS associates with leiomyoma in the Icelandic dataset (OR = 1.25, $P = 3.2 \times 10^{-55}$), which may be interpreted such that one standard deviation increase in the PGS leads to a 25% increased chance of leiomyoma. After correction for the number of phenotypes tested ($P < 0.05/29 = 1.72 \times 10^{-3}$), the PGS was also significantly correlated with the risk of being diagnosed with cancer (all cancer types together), thyroid cancer and prostate cancer (likelihood ratio test, Table 2).

## Discussion
Results from this meta-analysis highlight two distinct genetic pathways that play a role in the development of leiomyomas; one of genes linked to tumorigenesis (*TP53*, *TERT*, *ATM*, and *OBFC1*) and the other of variants and loci linked to hormone metabolism (*CDC42/WNT4*, *GREB1*, *MCM8*, and *SYN1/ESR1*). It is tempting to speculate that the tumor-associated variants provide genetic background for leiomyoma development, whereas the hormone-associated variants enhance the growth of the tumors, causing them to become symptomatic. A common genetic etiology between cancer and leiomyoma is further suggested by the leiomyoma PGS that associates with several cancers, including a phenotype consisting of individuals affected by any

type of cancer. We did not have the power to test the association of the variants with leiomyosarcoma—the malignant tumor originating in the myometrium— because of the rarity of this tumor type (44 cases in this study).

The association between leiomyoma and the 3′UTR variant in *TP53* is of particular interest. Contrary to most pathogenic mutations in *TP53*, the variant does not change the amino acid sequence of the protein, but instead disrupts normal polyadenylation and reduces the abundance of mRNA[11]. This results in a cancer risk spectrum quite distinct from the Li–Fraumeni tumor syndrome. Notably, the variant predisposes to some nonlethal tumor types such as basal cell skin cancer and colorectal adenomas, but not colorectal cancer[11]. The association between missense variants in *ATM* and leiomyoma is also unexpected. We have previously shown that loss-of-function mutations in *ATM* confer high risk of gastric cancer in the Icelandic population[45]. These same loss-of-function variants do not associate with leiomyoma risk, suggesting that the missense variants may have a mild effect on some specific functions of *ATM* in uterine tissue. Here, we report for the first time an association between the missense variant Asp1853Asn in *ATM* (rs1801516) and increased risk of squamous cell skin cancer.

In conclusion, we report the first GWAS of leiomyoma in European populations. Our results show a genetic overlap between leiomyoma and various cancers, and highlight the role of estrogen in tumor growth.

## Methods
**Datasets.** The meta-analysis combined the results of two GWAS of uterine leiomyoma. The Icelandic dataset consisted of 6,728 histologically confirmed cases of leiomyoma of the uterus (ICD codes ICD10 D25 and ICD9 218 and 219) diagnosed at the Department of Pathology, Landspitali University Hospital, compared to

124,542 female controls. The UK Biobank (UKB) leiomyoma dataset consisted of 9,867 cases with ICD10 D25 compared to 398,788 controls of both sexes. The UKB phenotype is based on diagnoses codes recorded for a participant during all visits to a hospital. Given the high prevalence of leiomyoma, it is clear that a fraction of the controls in both datasets will have undiagnosed leiomyoma. This under-diagnosis of the controls weakens the power of the GWAS, but should not lead to false-positive results.

The Icelandic cancer case–control datasets used were described previously[46]. The primary source of information on cancer is the Icelandic Cancer Registry (ICR), which has registered all solid, non-cutaneous cancers in the entire population of Iceland since 1955[47]. Registration of skin and hematological cancers started in 1980s. ICR registration is based on the ICD system and includes information on histology (systemized nomenclature of medicine, SNOMED). Over 94% of diagnoses in the ICR have histological confirmation. The endometriosis[32], bone phenotype[48], and age at menopause[49] datasets have been described previously. For sex-specific phenotypes in Iceland, only controls for the relevant sex were included in the analysis.

The study was approved by the Icelandic National Bioethics Committee (reference number. 17–124). Written informed consent was obtained from all genotyped subjects.

**Genotyping**. The Icelandic part of the study is based on the genotypes of 150,656 Icelanders, who have been genotyped using Illumina SNP chips along with whole-genome sequence (WGS) data from 15,220 Icelanders. The genotypes for the 150,656 individuals were long-range phased and imputed using information from the WGS individuals[50]. Using genealogic information, the sequence variants were imputed into 282,894 relatives of the genotyped individuals to further increase the sample size for association analysis and to increase the power to detect the associations[51]. A total of 34.5 million variants were used in the Icelandic GWAS. Datasets were constructed as a part of disease association efforts at deCODE genetics. For further information regarding genotyping and imputation, we refer to Gudbjartsson et al.[52]

Genotyping of UKB samples was performed using a custom-made Affymetrix chip, UK BiLEVE Axiom[53], and with the Affymetrix UK Biobank Axiom array[54]. Imputation was performed by the Wellcome Trust Centre for Human Genetics, using the Haplotype Reference Consortium (HRC) and the UK10K haplotype resources[55]. This yields a total of 96 million imputed variants, however only 27 million variants imputed using the HRC reference set passed the quality filters used in our study.

**GWAS and meta-analysis**. Logistic regression assuming an additive model was used to test for association between variants and disease, treating disease status as the response and expected genotype counts from imputation as covariates, and using the likelihood ratio test to compute $P$-values. The association analysis for both the Icelandic and UKB datasets was done using a software developed at deCODE genetics[52]. For the Icelandic study group, patients and controls are matched on gender and age at diagnosis or age at inclusion, and information on county of origin within Iceland are included as covariates to adjust for possible population stratification. For the UK datasets, cases and controls are restricted to individuals of genetically confirmed white British origin, and 40 principle components are included in the analysis to adjust for population substructure. To account for inflation in test statistics due to cryptic relatedness and stratification, we applied the method of linkage disequilibrium (LD) score regression[44] to estimate the inflation in test statistics and adjusted all P values accordingly. The estimated correction factor for Leiomyoma based on LD score regression was 1.13 for the Icelandic and 1.10 for the UK datasets, respectively.

Variants in the UKB imputation dataset were mapped to NCBI Build38 positions and matched to the variants in the Icelandic dataset based on allele variation. The results from the two cohorts were combined using a fixed-effect model, in which the cohorts were allowed to have different population frequencies for alleles and genotypes, but were assumed to have a common OR and were weighted with the inverse of the variance. We selected a threshold of 0.9 imputation info for variants available in the Icelandic dataset and 0.8 imputation info for variants only available in the UKB dataset. A total of 43,421,991 variants were used in the analysis. Heterogeneity was tested by comparing the null hypothesis of the effect being the same in all populations to the alternative hypothesis of each population having a different effect using a likelihood ratio test.

We used the weighted Holm–Bonferroni method[56] to account for all 43,421,991 variants being tested ($P$-value < (0.05×weight)/43,421,991). Using the weighing scheme given in Sveinbjornsson et al.[10], this procedure controls the family-wise error rate at 0.05; $P \leq 2.1 \times 10^{-7}$ for high-impact variants (including stop-gained and loss, frameshift, splice acceptor or donor and initiator codon variants, $n = 13,264$), $P \leq 5.0 \times 10^{-8}$ for missense, splice-region variants and in-frame-indels ($n = 257,401$), $P \leq 4.6 \times 10^{-9}$ for low-impact variants (including synonymous, 3′ and 5′ UTR, and upstream and downstream variants, $n = 3,303,568$), $P \leq 2.0 \times 10^{-9}$ for deep intronic and intergenic variants in DNase I hypersensitivity sites (DHS) ($n = 5,999,736$), and $P \leq 7.6 \times 10^{-10}$ for other non-DHS deep intronic and intergenic variants ($n = 33,998,502$).

**Conditional analysis**. We applied approximate conditional analyses, implemented in the GCTA software[57], to the meta-analysis summary statistics to look for additional association signals at each of the genome-wide significant loci. LD between variants was estimated using a set of 8700 whole-genome sequenced Icelandic individuals. The analysis was restricted to variants present in both the Icelandic and UKB datasets and within 1 Mb from the index variant. We tested 16 loci and about 30,000 variants in the conditional analysis and reported variants with conditional $P$ value $< 10^{-6}$. The results from GCTA were verified by conditional analysis using genotype data in the Icelandic and UK datasets separately, and the results presented in Table 1 are obtained by meta-analyzing those results.

**Polygenic score, heritability, and explained variance**. We derived a leiomyoma polygenic risk score (PGS) for the Icelandic individuals and, to avoid bias, we only used effect estimates from the GWAS summary results for leiomyoma from the UK dataset. The PGS was calculated using genotypes for about 630,000 autosomal markers included on the Illumina SNP chips to avoid uncertainty due to imputation quality. We estimated linkage disequilibrium (LD) between markers using 14,938 phased Icelandic samples and used this LD informaton to calculate adjusted effect estimates using LDpred[58,59]. We created several PGSs assuming different fractions of causal markers (the P parameter in Ldpred) and selected the best one based on the prediction of leiomyoma in the Icelandic dataset (0.3% causal variants). This PGS was then used when we calculate the correlation of the PGS with other phenotypes in the Icelandic dataset.

We computed LD scores for genotyped individuals in the Icelandic cohort using only high-quality markers and estimated the heritability explained by all markers, with MAF above 1% using LD score regression[44]. We estimated the risk ratio among siblings ($\lambda_S$) of Icelandic leiomyoma cases ($N = 6728$) by cross-matching with a genealogy database that covers the entire Icelandic nation. The risk ratio among siblings ($\lambda_S$) was estimated at 1.74 [1.63, 1.85] ($P < 0.0001$) using an approach previously described by Edvardsson et al[60]. This allows us to calculate the proportion of sibling recurrence risk of leiomyoma explained by the signals identified in the present study, $\log(\lambda_{S[i]})/\log(\lambda_S)$[61].

**Functional annotation of leiomyoma variants**. Variants in linkage disequilibrium (LD) with the lead variants were identified on the basis of in-house genotype data using $r^2 > 0.8$ for pairwise comparison of the nearest 100,000 variants to define an LD class. These variants were then annotated by intersection with chromatin immunoprecipitation (ChIP) signal data for uterine tissue. The ChIP-seq data was derived from the ENCODE project (www.encodeproject.org) downloaded in preprocessed (MACS v2 algorithm) bigWig format representing analysis of acetylation of lysine K27 of histone H3 (H3K27ac) with accession number ENCFF407ITR. The signal $P$-values were adjusted by the Benjamini-Hochberg procedure to account for multiple hypothesis and thresholded at the 1% FDR significance level. DNase hypersensitivity data (DHS) for uterus tissue were also downloaded in preprocessed format, accession number ENCFF604WBU, and intersected with variant position in each LD class.

Chromatin interaction map data were derived from Hi-C sequencing for various cell types and primary tissue samples[62]. The data were downloaded from Omnibus, accession number GSE87112, in pre-processed format (Fit-Hi-C algorithm) representing false-discovery rates (FDR) for contact regions at 40 kb resolution. To define statistically significant contacts we used a threshold value of FDR $< 10^{-6}$ in mesoendoderm cells[62]. We confined this analysis to variants residing within candidate regulatory regions based on Encode data for H3K27ac and DHS in uterus tissue. The regions found to interact with the variant-containing regions were then queried for Refseq genes and intersected with a database for cancer genes (NCG 5.0; ncg.kcl.ac.uk) and tissue-specific expression patterns as provided in the Human Protein Atlas (www.proteinatlas.org), v18 downloaded in December 2017.

## Data availability

The Icelandic population WGS data has been deposited at the European Variant Archive under accession code PRJEB8636. The authors declare that the data supporting the findings of this study are available within the article, its Supplementary Data files and upon request. The UK Biobank data can be obtained upon application (ukbiobank.ac.uk).

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

## Acknowledgements

This research has been conducted using the UK Biobank Resource under Application Number '24711'. We thank the individuals who participated in the study and whose contribution made this work possible. We acknowledge the Icelandic Cancer Registry for assistance in the ascertainment of the cancer patients.

## Author contributions

T.R., B.G., P.S., B.V.H., G.T, and K.S. designed the study and interpreted the results. A.S, A.T., E.A.H., K.K., O.I., V.G., R.T.G., R.A., J.G.J., and K.O. carried out the subject ascertainment, recruitment, and collection of clinical data. U.S., V.S., S.N.S., J.G., G.A.A., A.O., F.Z., G.H., R.P.K., O.B.D., G.M., V.T., F.W.A., and U.T. collected, processed, and analyzed the genotype and phenotype data. O.A.S. performed the functional annotations. A.I., M.L.F., L.S., J.K.S., G.S., D.F.G., B.G., and B.V.H., performed the statistical and bioinformatics analyses. T.R., B.G., O.A.S, G.T., and K.S., drafted the manuscript. All authors contributed to the final version of the paper.

## Additional information

**Competing interests:** The authors that are affiliated with deCODE are employees of deCODE genetics/Amgen. The remaining authors declare no competing interests.

