## [Peer Review File · Nature Communications]

Reviewers' comments:

Reviewer #1 (Remarks to the Author):

This is a meta analysis study of two genome-wide association studies from Iceland and the UK. From 16,595 cases and 523,330 controls, 19 variants were associated with leiomyomas. This study revealed GWAS hits that are highlighted into genetic 2 pathways: cancer and hormones. Novel hits were identified as well as confirmation with hits with the Japanese GWAS study. The identification of the 2 potential pathways that are highlighted through GWAS analysis is new. This however, is not conceptually not surprising, as leiomyomas do exhibit characteristics of growing tumors and they are hormone responsive tumors. The identification and potential function of the 19 variants are exciting and new information is provided.

Reviewer #2 (Remarks to the Author):

This is an interesting study which provides results of interest to the clinical and research community regarding the relationship between "benign" conditions and cancer risk. In addition to providing evidence for multiple highly significant associations of genetic loci with leiomyoma (some of which are direct replication of loci identified by a Japanese GWAS, providing confidence in the meta-analysis), the authors have explored likely target genes for functional follow-up. The authors have also utilized the availability of additional phenotype data from the Icelandic dataset to explore risk associations beyond leiomyoma.

Some specific comments are provided below.

Abstract

It would be helpful to state which malignant/benign tumors have been associated with the 5 variants at four loci.

Can the authors specifically comment on the overlap between UL loci designated as previously discovered for endometriosis and hormone-related traits, and known cancer risk loci – *esr1* is stated as a hormone-related trait (which it obviously is), but GWAS have definitely implicated *ESR1* in breast cancer risk. Is this because the authors are only referring to cancer or hormonal traits picked up in their Icelandic comparison? It would seem so (after later reading the result section).

Clarification of the basis for designating loci is important.

Regarding the loci previously discovered by the Japanese study – it would appear from the text that there is strong correlation of European hits for 2 of 4 Japanese loci? Recommend removing this from the abstract (where the suggestion is that 3 snps are directly replicated) and explain in more detail in the text, where there can be some discussion about LD in the region for the different populations.

In the abstract please explain better how polygenic score and cancer risk was tested – it is

unclear that this was in the Icelandic population only – with implications for sample size and bias (although that is addressed in the methods).

Confirm upfront in abstract and results (somewhere in lines 77-80) that this study was a GWAS of women of European ancestry – genetically confirmed for UKBB samples at least - relevant because of the comparison to Japanese results.

Results/Discussion

Line 81 describe further the rationale for weighting the significance thresholds.

Importantly, the authors should also present the data for all SNPs considered significantly associated using conventional GWAS significance thresholds- this will allow comparison of results to preceding unweighted GWAS of hormone-related traits/diseases, and allow consideration of the data using different weighting schemes that include additional biological data or other factors.

Table 1 should be updated to include the references for the other phenotypes, and it should be clarified in the text if all previously reported associations were at GWS or not – and if not, then those specifically noted as such.

Can the authors comment on the R² between the SYNE1/ESR1 hit and the various SNPs that have been reported to be associated with different breast cancer phenotypes.

Although none of the 5 cancer variants affects risk of endometrial cancer in Iceland, can the authors comment on the overlap with results from larger-scale endometrial cancer studies. This is relevant to ESR1 and TERT as least, where there are some findings for these loci, albeit at sub-genome-wide significance levels.

For the discussion of identification of risk SNPs at loci highlighted by the previous Japanese GWAS, please include some information about differences in LD structure between the groups. Likewise a discussion on LD around risk locus previously identified in African American women (22q13.1, CYTH4), and not identified in this study of European women would be helpful.

Line 145 – as written it is difficult to follow – Sup Table 6 presents the risk estimates derived from the leiomyoma GWAS (assume only Icelandic – right? – since presented with endometriosis excluded and there is no indication that this sub-analysis was done for the UKBB).

Table 2 – are there any genes in italics (as per footnote 3). ESR1 was pulled out in the footnote but is not in italics. Perhaps just list this in the footnote (there was only one in bold too)

Table 3 – please clarify in the title that the risk analysis includes Icelandic samples only.

And clarify the difference in sample sizes to what is in sup T4?

Given that obesity and estrogen levels are established risk factors for uterine leiomyomas, and large-scale GWAS results for these traits, it would be beneficial for the authors to consider mendelian randomization analysis to assess the contribution of these traits to leiomyoma, and to justify the last statement of the conclusion relating to the role of estrogen in tumor growth.

Minor comments

Online methods

-line 223/224 some words are missing – the sentence does not make sense.

-line 237 – imputed should read imputed

-line 265 – inverse should read inverse

-line 282 – typo uncertainty should be uncertainty

-line 284 – remove “the” between “different” and “fractions”

-line 285/286 – grammar needs some work

Other

Several other typographical errors, including

Title of sup Table 1

Sub-title of sup table 4

Legend to Figure 2

Reviewer #3 (Remarks to the Author):

This manuscript describes a genomewide association study using data from Iceland and from UK Biobank to look for common genetic variants associated with risk of leiomyoma (fibroids). This is the first leiomyoma GWAS in a European population. Nineteen independent genetic variants (16 loci) were identified as having associations with leiomyoma which were significant at the GWAS threshold, several of which are at loci containing known associations with cancer, endometriosis or hormonal traits.

The study follows a standard GWAS format, and appears to have been carried out in a competent manner. The results will be of interest to those involved in research around uterine conditions, and to those who are interested in the identified candidate causal genes because of their associations with other traits.

The introduction notes that around 70% of Caucasian women over the age of 50 years have a leiomyoma, although the majority of cases are asymptomatic. However, both studies contained far more controls than cases (9,867 cases to 398,788 controls in UK Biobank). Around half the Biobank cohort is female, mostly over 50, so there should be very many more than ~10,000 cases. Presumably this is because only a minority of cases are ever diagnosed – hence many of the ‘controls’ will actually be cases, and it would be more accurate to describe the leiomyoma cases as cases which were severe enough to require hospitalization (the cases appear to have been identified via hospital episode data - is this correct, or was information about health conditions given by participants at the registration

interview included?). Were any exclusion criteria used for the controls, beyond standard QC. Similar issues may apply to the Icelandic data. This should be clarified, and the implications of under-diagnoses should be addressed in the text.

The text notes that 3 of the additional hits at 5 loci reached genome-wide significance i.e. the second signals at KANK1/DMRT1, BETL1/SIRT3 and ATM shown in Table 1. However, Sup Table 3 shows the second signal at the GREB2 locus as having conditional $p < 5e-8$, whereas the second signal at ATM does not meet this threshold in the conditional analysis. This should be clarified, and the ORs and p-values in Table 1 for the multi-SNP loci should be replaced with those from the conditional analyses.

Imputation information scores should be included for all the SNPs in Table 1 to show the quality of the imputation for SNPs which were not genotyped.

Sup Fig 1 (the Locus Zoom plots) would be easier to follow if the plots were in the same order as the SNPs in Table 1, or at least if the pairs of plots relating to the same locus were displayed next to one another. The plot for rs16991615 (MCM8, chr 20) shows that no SNPs are in moderate/strong LD with the top SNP – it is particularly important to check the quality of the imputation for this SNP (if it was not directly genotyped). Similarly for the LINC0 SNP (chr 13).

It is good to know that two of the previously reported leiomyoma variants were replicated here, but intriguing that the best SNP in the OBFC1 region in this study is not in LD with the best SNP at this locus in the Japanese study. What was the p-value for the reported Japanese variant in this study?

Testing the association of the SNPs with leiomyoma after excluding endometriosis is a good check, and it is reassuring that the associations remained in this subset. However, could the authors comment on how effective this will have been in removing all endometriosis cases, given that this is also known to be an underdiagnosed condition?

How were the 28 tumor/hormone-related phenotypes in Table 3 selected? The numbers of cases in Table 3 are very different from those in Sup Table 4, which lists e.g. 44 leiomyosarcomas, 6,013 breast cancers, 42,331 all-cancers and 10,216 women with menopause data. Which numbers are correct?

The correction factor of 10.1 for the UK dataset (p12) must be a typo?

Manuscript Number NCOMMS-18-06092-T

Variants associating with uterine leiomyoma highlight genetic background shared by various cancers and hormone-related traits.

Point-by-point response to reviewers' comments.

Reviewer #1 (Remarks to the Author):

This is a meta analysis study of two genome-wide association studies from Iceland and the UK. From 16,595 cases and 523,330 controls, 19 variants were associated with leiomyomas. This study revealed GWAS hits that are highlighted into genetic 2 pathways: cancer and hormones. Novel hits were identified as well as confirmation with hits with the Japanese GWAS study. The identification of the 2 potential pathways that are highlighted through GWAS analysis is new. This however, is not conceptually not surprising, as leiomyomas do exhibit characteristics of growing tumors and they are hormone responsive tumors. The identification and potential function of the 19 variants are exciting and new information is provided.

Response: *We are pleased that the reviewer shares our enthusiasm for the results of our study.*

Reviewer #2 (Remarks to the Author):

This is an interesting study which provides results of interest to the clinical and research community regarding the relationship between "benign" conditions and cancer risk. In addition to providing evidence for multiple highly significant associations of genetic loci with leiomyoma (some of which are direct replication of loci identified by a Japanese GWAS, providing confidence in the meta-analysis), the authors have explored likely target genes for functional follow-up. The authors have also utilized the availability of additional phenotype data from the Icelandic dataset to explore risk associations beyond leiomyoma.

Some specific comments are provided below.

Abstract

It would be helpful to state which malignant/benign tumors have been associated with the 5 variants at four loci.

Response: *We agree that this information would be of interests. However, the number of tumor types is too large to fit within the word limits of the abstract. In table 1 we name 14 tumor types (benign and malignant) that have been associated with the 5 variants. To give some feeling of the multitude of tumor types, we have added the word "various" to the following sentence: "Five of the variants were previously reported to confer risk of **various** malignant or benign tumors (rs78378222 in TP53, rs10069690 in TERT, rs1800057 and rs1801516 in ATM, rs7907606 at OBFC1)..."*

Can the authors specifically comment on the overlap between UL loci designated as previously discovered for endometriosis and hormone-related traits, and known cancer risk loci – *esr1* is stated as a hormone-related trait (which it obviously is), but GWAS have definitely implicated *ESR1* in breast cancer risk. Is this because the authors are only referring to cancer or hormonal traits picked up in their Icelandic comparison? It would seem so (after later reading the result section).

Clarification of the basis for designating loci is important.

Response: *There is indeed a complex overlap between the 4 hormone-related traits loci and cancer risk. First, at the *ESR1* locus multiple variants have been reported that associate with various types of breast cancer, however, none of these variants is strongly correlated to our UL variant. Second, one of the two leiomyoma variants in *GREB1* is moderately correlated with a reported breast cancer variant ($r^2=0.44$). Third, the variant in *MCM8* has also been reported to associate with breast cancer. Fourth, at the *CDC42/WNT4* locus, variants that associate with risk of ovarian cancer have been reported but no variants for breast cancer.*

*The breast cancer associations at *MCM8* and *GREB1* had escaped us and we have added this information to table 1 and the main text with the appropriate references.*

We have changed the sentence in the abstract to: "... and four signals are located at established risk loci for hormone-related traits such as endometriosis and breast cancer, at 1q36.12...."

Regarding the loci previously discovered by the Japanese study – it would appear from the text that there is strong correlation of European hits for 2 of 4 Japanese loci? Recommend removing this from the abstract (where the suggestion is that 3 snps are directly replicated) and explain in more detail in the text, where there can be some discussion about LD in the region for the different populations.

Response: *According to the reviewer's suggestion, we have removed the sentence from the abstract.*

*To address this issue, we looked at the correlation between the two variants, *rs7907606* (variant reported in our meta-analysis) and *rs7913069* (reported in Japan), in the 1000 genomes dataset and note that the two variants are not correlated, neither in Caucasians ($r^2 = 0.01$) nor in East-Asians ($r^2 < 0.01$).*

*We then tested 278 variants that correlate with *rs7913069* (with $r^2 > 0.1$) in the Japanese population for association with leiomyoma in our dataset. The lowest P-value observed is 0.00015 for *rs78381949*, a variant that correlates with $r^2 = 0.23$ with *rs7913069* in East-Asians but has very little correlation with *rs7907606* ($r^2 = 0.04$) in Caucasians. It is thus unlikely that *rs7907606* and *rs7913069* could be tagging the same association signal. We have added this information to the Results section.*

*Our results are consistent with the results of Edwards et al (ref 8 in the manuscript) which replicated the same two variants as we do but not *rs7913069**

In the abstract please explain better how polygenic score and cancer risk was tested – it is unclear that this was in the Icelandic population only – with implications for sample size and bias (although that is addressed in the methods).

Response: *The abstract now states that the polygenic risk score was computed using the UKB leiomyoma data and tested for cancer in the Icelandic population.*

Confirm upfront in abstract and results (somewhere in lines 77-80) that this study was a GWAS of women of European ancestry – genetically confirmed for UKBB samples at least - relevant because of the comparison to Japanese results.

Response: *This point has been added to the abstract and the first paragraph of the results.*

Results/Discussion

Line 81 describe further the rationale for weighting the significance thresholds.

Response: *We have added a sentence to the Results section that refers to the rationale for selecting GWS thresholds and added a description of the weighing scheme and information on the number of variants in each annotation class to the Methods section. We have also added the number of variants used in the genetic analysis which was not included in the original submission.*

*To clarify, we do not use the flat genome-wide significance P-value threshold of 5.0×10^{-8} that has become the standard for GWAS's and accounts only for 1M tests. We use a weighted Bonferroni correction that accounts for all 43M variants tested. Each hypothesis is rejected if the P-value < $(0.05 * \text{weight}) / 43,421,991$ and using the weights given in Sveinbjornsson et al. this procedure controls the family-wise error rate at 0.05. Being similar to normal Bonferroni correction, this procedure is conservative when tests are not independent making these thresholds stringent due to linkage disequilibrium between variants.*

Importantly, the authors should also present the data for all SNPs considered significantly associated using conventional GWAS significance thresholds- this will allow comparison of results to preceding unweighted GWAS of hormone-related traits/diseases, and allow consideration of the data using different weighting schemes that include additional biological data or other factors.

Response: *We have added a new supplementary table (Sup table 3) with results of all variants with P value < 5.0E-8.*

Table 1 should be updated to include the references for the other phenotypes, and it should be clarified in the text if all previously reported associations were at GWS or not – and if not, then those specifically noted as such.

Response: *We have added references to table 1 as requested by the reviewer and added a sentence in the footnote to Table 1 stating that all the reported associations referred to in the comment column were reported to be GWS.*

Can the authors comment on the R2 between the SYNE1/ESR1 hit and the various SNPs that have been reported to be associated with different breast cancer phenotypes.

Response: *As noted in the response to the second comment of reviewer 2, the ESR1 locus contains multiple variants that have been reported to associate with various types of breast cancer in different ethnicities. None of these variants is strongly correlated to our UL variant and we have added a sentence on this observation to the Results section, highlighting 3 breast cancer variants confirmed in European women.*

Although none of the 5 cancer variants affects risk of endometrial cancer in Iceland, can the authors comment on the overlap with results from larger-scale endometrial cancer studies. This is relevant to ESR1 and TERT as least, where there are some findings for these loci, albeit at sub-genome-wide significance levels.

Response: *To address the reviewer's point, we added data on endometrial cancer from the UKB. We assessed the association between the leiomyoma variants and endometrial cancer using a meta-analysis of Icelandic and UKB data, increasing the number of cases and controls to 2,462 and 336,978, respectively. Only rs10917151 (CDC42/WNT4) was significantly associated with endometrial cancer ($P=0.00045$, OR 1.14) after correcting for the number of tests ($P=0.05/21 = 0.0024$). We have added a supplementary table with the results (**Sup table 8**). In addition, we tested the association between leiomyoma and 7 reported endometrial cancer risk variants and present these results in a new supplementary table (**Sup table 9**). None of the endometrial cancer variants show significant association with leiomyoma.*

For the discussion of identification of risk SNPs at loci highlighted by the previous Japanese GWAS, please include some information about differences in LD structure between the groups. Likewise a discussion on LD around risk locus previously identified in African American women (22q13.1, CYTH4), and not identified in this study of European women would be helpful.

Response: *See response to comment 3 of reviewer 2 above regarding the variants at the OBFC1 locus. The variant at the CYTH4 locus identified in African American women, rs739187, does not associate with UL in our data ($P=0.51$, OR=1.01). rs739187 is about 3 Mb away from the UL variant rs12484951 (TNRC6B) and the 2 variants are not correlated either in Europeans ($r^2=0.0064$) or African Americans ($r^2=0.017$). We have added this information to the Results section.*

Line 145 – as written it is difficult to follow – Sup Table 6 presents the risk estimates derived from the leiomyoma GWAS (assume only Icelandic – right? – since presented with endometriosis excluded and there is no indication that this sub-analysis was done for the UKBB).

Response: *Indeed this analysis was only done in Iceland. We have rewritten this paragraph to make this fact clearer.*

Table 2 – are there any genes in italics (as per footnote 3). ESR1 was pulled out in the footnote but is not in italics. Perhaps just list this in the footnote (there was only one in bold too)

Response: *It seems that while transferring Table 2 to Word format, some of the formatting was lost. In total, 8 genes should have been bolded and 2 genes put in italics. We have now corrected this and apologize for the confusion.*

Table 3 – please clarify in the title that the risk analysis includes Icelandic samples only. And clarify the difference in sample sizes to what is in sup T4?

Response: *We have changed the title of Table 3 as suggested. We also state that the Polygenic score was calculated only for chip-genotyped individuals.*

The discrepancy in numbers stems from the fact that the GWAS analysis (Sup Table 4) includes both chip-genotyped and familiarly imputed cases and controls whereas the polygenic risk score presented in Table 3 is based only on chip-typed individuals. The number of individuals used in the risk score analysis is correct in Table 3. To clarify this difference in numbers, we have added text to Methods, Results and legend to Sup. table 4. In particular, we have removed the referral to the risk score analysis in the legend to Sup table 4 which was wrong.

Given that obesity and estrogen levels are established risk factors for uterine leiomyomas, and large-scale GWAS results for these traits, it would be beneficial for the authors to consider mendelian randomization analysis to assess the contribution of these traits to leiomyoma, and to justify the last statement of the conclusion relating to the role of estrogen in tumor growth.

Response: *To get some insight into this issue, we have tested the association between a genetic risk score for BMI, based on the meta-analysis done by the GIANT consortium (excluding contribution from Icelandic samples), and leiomyoma in the Icelandic dataset. We do not observe any correlation between the risk score and leiomyoma ($P = 0.53$), which does not support the idea that obesity, or more precisely variation in BMI, is a causal factor for leiomyoma. We note that the BMI risk score used is well powered to detect such association; it is based on effect estimates for about 600,000 variants and in a sample size of 300,000 individuals and shows strong correlation with many other traits for which BMI is a plausible causal factor. The same is observed if we look at the strongest BMI associated variant, in the FTO gene, which associates with BMI with $P = 3e-62$ in the Icelandic data, but does not association with leiomyoma ($P = 0.19$). We do, however, not feel comfortable making statements on this issues in the current manuscript.*

Minor comments

Online methods

-line 223/224 some words are missing – the sentence does not make sense.

- line 237 – imputed should read imputed
- line265 – inverse should read inverse
- line282 – typo uncertainty should be uncertainty
- line 284 – remove “the” between “different” and “fractions”
- line 285/286 – grammar needs some work

Other

Several other typographical errors, Including

Title of sup Table 1

Sub-title of sup table 4

Legend to Figure 2

Response: *We have corrected the typographical errors and grammar in the locations pointed out by the reviewer.*

Reviewer #3 (Remarks to the Author):

This manuscript describes a genomewide association study using data from Iceland and from UK Biobank to look for common genetic variants associated with risk of leiomyoma (fibroids). This is the first leiomyoma GWAS in a European population. Nineteen independent genetic variants (16 loci) were identified as having associations with leiomyoma which were significant at the GWAS threshold, several of which are at loci containing known associations with cancer, endometriosis or hormonal traits.

The study follows a standard GWAS format, and appears to have been carried out in a competent manner. The results will be of interest to those involved in research around uterine conditions, and to those who are interested in the identified candidate causal genes because of their associations with other traits.

The introduction notes that around 70% of Caucasian women over the age of 50 years have a leiomyoma, although the majority of cases are asymptomatic. However, both studies contained far more controls than cases (9,867 cases to 398,788 controls in UK Biobank). Around half the Biobank cohort is female, mostly over 50, so there should be very many more than ~10,000 cases. Presumably this is because only a minority of cases are ever diagnosed – hence many of the ‘controls’ will actually be cases, and it would be more accurate to describe the leiomyoma cases as cases which were severe enough to require hospitalization (the cases appear to have been identified via hospital episode data - is this correct, or was information about health conditions given by participants at the registration interview included?). Were any exclusion criteria used for the controls, beyond standard QC. Similar issues may

apply to the Icelandic data. This should be clarified, and the implications of under-diagnoses should be addressed in the text.

Response: *This is an important point. As pointed out by the reviewer, a fraction of the controls will have un-diagnosed leiomyoma. However, while this weakens the power of the GWAS, it should not lead to false-positive results. We have added this note to the description of the study datasets in the Methods section. In our study, there were no exclusions criteria used for controls other than the QC filters described in the Methods section.*

All cases, both from Iceland and UKB, are hospital-based and have histologically confirmed diagnoses with ICD10 code D25. There are no self-reported cases in the study. We have added text to the first paragraph of the Results to emphasize this. Although a large fraction of hysterectomies are due to leiomyoma, other pathologies will lead to diagnosis of leiomyoma, such as endometriosis, uterine prolapse and cancer. It might not even be unreasonable to say that the present GWAS reflects uterine problems leading to a hysterectomy. Therefore it is very valuable to be able to test associations of the variants to these other, confounding pathologies in the current study. We have modified and added to the text where we assess the overlap between leiomyoma, endometriosis and endometrial cancer (top of page 7 in the current version).

The text notes that 3 of the additional hits at 5 loci reached genome-wide significance i.e. the second signals at KANK1/DMRT1, BETL1/SIRT3 and ATM shown in Table 1. However, Sup Table 3 shows the second signal at the GREB2 locus as having conditional $p < 5e-8$, whereas the second signal at ATM does not meet this threshold in the conditional analysis. This should be clarified, and the ORs and p-values in Table 1 for the multi-SNP loci should be replaced with those from the conditional analyses.

Response: *We are grateful to the reviewer for pointing out this discrepancy in data presentation. We have updated Table 1 to include both the GW-significant signals and the secondary signals that were significant in the conditional analysis (testing approximately 30,000 markers, Bonferroni corrected $P < 1.0E-6$). The P values from the conditional analysis are presented and the secondary signals are marked with a superscript and noted as such. Table 1 now contains 21 variants. Consequently, we have updated the whole manuscript (text, tables and figures), taking all 21 variants into accounts.*

Imputation information scores should be included for all the SNPs in Table 1 to show the quality of the imputation for SNPs which were not genotyped.

Response: *The imputation information for all variants in Iceland and UKB has been added to Table 1.*

Sup Fig 1 (the Locus Zoom plots) would be easier to follow if the plots were in the same order as the SNPs in Table 1, or at least if the pairs of plots relating to the same locus were displayed next to one another. The plot for rs16991615 (MCM8, chr 20) shows that no SNPs are in moderate/strong LD with the top SNP – it is particularly important to check the quality of the imputation for this SNP (if it was not directly genotyped). Similarly for the LINCO SNP (chr 13).

Response: *The locusplots have been ordered according to chromosome location consistent with Table 1. Also, we added the imputation information to Table 1 as mentioned the comment above.*

It is good to know that two of the previously reported leiomyoma variants were replicated here, but intriguing that the best SNP in the OBFC1 region in this study is not in LD with the best SNP at this locus in the Japanese study. What was the p-value for the reported Japanese variant in this study?

Response: *The association results in our data for the variant reported in the Japanese population, rs7913069, were as follows: $P=0.98$, $OR=1.0$, imputation information in both Iceland and UKB is 1.00. We have added the association numbers for the Japanese variants to the Results. Our results are consistent with the results of Edwards et al (ref 8 in the manuscript) which replicated the same two variants as we do but not rs7913069. See also response to comment 3 of reviewer 2.*

Testing the association of the SNPs with leiomyoma after excluding endometriosis is a good check, and it is reassuring that the associations remained in this subset. However, could the authors comment on how effective this will have been in removing all endometriosis cases, given that this is also known to be an underdiagnosed condition?

Response: *Our endometriosis case group includes 1857 cases of whom 695 have severe disease (stage III or IV). All of the cases were surgically diagnosed and will therefore represent the fraction of cases with a more severe disease. As the reviewer points out, there is a high likelihood that there will be some endometriosis cases left in the excluded group and we have added a sentence to the Results to alert the reader to this fact.*

How were the 28 tumor/hormone-related phenotypes in Table 3 selected? The numbers of cases in Table 3 are very different from those in Sup Table 4, which lists e.g. 44 leiomyosarcomas, 6,013 breast cancers, 42,331 all-cancers and 10,216 women with menopause data. Which numbers are correct?

Response: *The cancer phenotypes were selected because they were 1) the most common cancer types or 2) relevant to this study (e.g. leiomyosarcoma). The hormone-related phenotypes were selected because they have good power in the Icelandic population and have been used in large GWAS studies.*

As regards the number of cases in the two kinds of analysis, both numbers are correct. The discrepancy in numbers stems from the fact that the GWAS analysis (Sup Table 4) includes both chip-genotyped and familiarly imputed cases and controls whereas the polygenic risk score presented in Table 3 is based only on chip-typed individuals. Please see response to the comment on Table 3 made by reviewer 2 on how we corrected this confusion.

The correction factor of 10.1 for the UK dataset (p12) must be a typo?

Response: *Indeed! The correction factor for UKB is 1.10, this has been corrected.*

Note from the authors:

In addition to the changes listed above, we calculated the sibling recurrence risk for leiomyoma and add an estimate of the proportion of sibling recurrence risk for leiomyoma explained by the 21 variants.

REVIEWERS' COMMENTS:

Reviewer #1 (Remarks to the Author):

The revision is satisfactory

Reviewer #2 (Remarks to the Author):

The authors have addressed the comments.

I still think it would be helpful to include the MR of obesity in the manuscript, given that the results are really informative and contradict statements about known risk factors that the authors are putting forward.

Reviewer #3 (Remarks to the Author):

I am satisfied that the authors have addressed all of the comments which I had previously made. I only have one remaining comment: The list of independent breast cancer SNPs listed on pages 6-7 is not quite complete – Dunning et al 2016 (PMID: 26928228) found five independent signals, tagged by rs3757322, rs9397437, rs851984, rs9918437, rs2747652.

Manuscript Number NCOMMS-18-06092-A

Variants associating with uterine leiomyoma highlight genetic background shared by various cancers and hormone-related traits

Point-by-point response to reviewers' comments.

Reviewer #1 (Remarks to the Author):

The revision is satisfactory

Reviewer #2 (Remarks to the Author):

The authors have addressed the comments.

I still think it would be helpful to include the MR of obesity in the manuscript, given that the results are really informative and contradict statements about known risk factors that the authors are putting forward.

Response: *We agree that the results of the analysis of obesity and leiomyoma that we did to address the issue raised by the reviewer are interesting. However, these results were preliminary and since they go against the current thought that obesity is a risk factor for leiomyoma, we feel that a much more thorough and extensive analysis would have to be done before publishing. We therefore chose not to include it in the current manuscript.*

Reviewer #3 (Remarks to the Author):

I am satisfied that the authors have addressed all of the comments which I had previously made. I only have one remaining comment: The list of independent breast cancer SNPs listed on pages 6-7 is not quite complete – Dunning et al 2016 (PMID: 26928228) found five independent signals, tagged by rs3757322, rs9397437, rs851984, rs9918437, rs2747652.

Response: *We are grateful to the reviewer for pointing out our omission of the paper by Dunning et al. In the current version of the manuscript, we include a total of 6 independent breast cancer signals published in three publications (Stacey et al, Dunning et al and Michailidou et al). In all cases the r^2 between the lead leiomyoma variant and the breast cancer variants is less than 0.03.*